# Labdane Diterpenoids from *Leonotis ocymifolia* with Selective Cytotoxic Activity Against HCC70 Breast Cancer Cell Line

**DOI:** 10.3390/diseases13050140

**Published:** 2025-05-01

**Authors:** Jane Busisiwe Ncongwane, Vuyelwa Jacqueline Tembu, Comfort Mduduzi Nkambule, Douglas Kemboi, Gerda Fouche, Nyeleti Vukea, Jo-Anne de la Mare

**Affiliations:** 1Department of Chemistry, Tshwane University of Technology, Private Bag X680, Pretoria 0001, South Africa; ncongwanejb@tut.ac.za (J.B.N.); nkambulecm@tut.ac.za (C.M.N.); 2School of Science and Technology, University of Kabianga, Kericho 2030-20200, Kenya; 3Department of Chemistry, University of Pretoria, Pretoria 0001, South Africa; gerda.fouche@up.ac.za; 4Department of Biochemistry and Microbiology, Rhodes University, Grahamstown 6140, South Africa; nyeletivukea@gmail.com (N.V.); j.delamare@ru.ac.za (J.-A.d.l.M.)

**Keywords:** *Leonotis*, *Leonotis ocymifolia*, labdane diterpenoids, *bis*-spirolabdane, MCF-7, HCC70, cytotoxic activity, breast cancer

## Abstract

Background: Triple-negative breast cancer (TNBC) is an aggressive subtype of breast cancer with limited therapeutic options. *Leonotis ocymifolia* is a shrub widely used in traditional medicine to alleviate cancer-related symptoms. In a search to find safe and efficacious therapeutic agents from medicinal plants, *Leonotis ocymifolia* was studied to find compounds with anticancer activity against TNBC. Methods: Compounds from *Leonotis ocymifolia* were characterized using spectroscopic data such as IR, 1D and 2D NMR, and MS spectrometry and evaluated for cytotoxic activity against triple-negative breast cancer (HCC70), hormone receptor-positive breast cancer (MCF-7), and non-tumorigenic mammary epithelial cell lines (MCF-12A). Results: A previously unreported *bis*-spirolabdane, 13*S*-nepetaefolin (**1**), together with five known labdane diterpenoids, namely nepetaefolin (**2**), dubiin (**3**), nepetaefuran (**4**), leonotin (**5**), and leonotinin (**6**), from the genus *Leonotis* were isolated. Overall, the labdane diterpenoids showed selective activity toward triple-negative breast cancer cells (HCC70). Of the compounds extracted, 13*S*-nepetaefolin demonstrated the greatest cytotoxic activity with an IC_50_ of 24.65 µM (SI = 1.08) against HCC70 cells; however, it was equally cytotoxic to non-tumorigenic MCF-12A breast cells (IC_50_ of 26.55 µM), whereas its isomer (**2**) showed no activity. This suggests that stereochemistry might have an effect on the cytotoxic activity of the *bis*-spirolabdane diterpenoids. All the compounds (**1**–**6**) demonstrated adsorption, distribution, metabolism, and excretion properties (ADME), while leonotin (**5**) and leonotinin (**6**) exhibited lead-like properties and high synthetic accessibility scores. Conclusions: The findings from this study warrant further investigation of *L. ocymifolia* for potential triple-negative breast cancer (TNBC) therapeutic agents, including potential chemical derivatization of *bis*-spiro labdane diterpenoid (**1**) to improve selectivity to TNBC over non-cancer cells.

## 1. Introduction

Globally, breast cancer ranks as the second most common cancer overall and is the most often diagnosed disease among women [1]. Triple-negative breast cancer (TNBC) is more common among women of African heritage and Asians compared to other breast cancer subtypes [2]. Triple-negative breast cancer (TNBC) is molecularly defined by the absence of expression of three prognostically significant receptors: the progesterone receptor (PR), estrogen receptor (ER), and human epidermal growth factor receptor 2 (HER2) [3]. Despite constituting about 10–15% of all breast cancers, triple-negative breast cancer (TNBC) is of concern due to its earlier onset, high mortality rate [4], high metastatic potential, and low survival rate [1].

Targeted therapy for breast cancer, including tamoxifen and trastuzumab, are utilized for ER/PR-positive and HER2-positive breast cancers, respectively [5,6]; however, they are ineffective against triple-negative breast cancer (TNBC) due to the lack of these receptors. The treatment of triple-negative breast cancer (TNBC) with traditional chemotherapy agents, including anthracyclines and taxanes, yields low therapeutic efficacy due to non-selective toxicity and the development of drug resistance [7]. Consequently, in search of safe and efficacious therapeutic agents for triple-negative breast cancer (TNBC) derived from medicinal plants, *Leonotis ocymifolia* was investigated owing to its documented traditional application in alleviating cancer-related symptoms and the presence of bioactive compounds [8].

*Leonotis ocymifolia* (Figure 1) is a plant used in traditional medicine for the treatment of various conditions, including diarrhea, wounds, stomach pain, headache, hypertension, malaria, asthma, diabetes, and dermatitis [9,10,11,12].

Plant extracts and essential oils demonstrate various pharmacological activities, including nephroprotective, analgesic, anti-inflammatory, antibacterial, anthelmintic, antiprotozoal, antimicrobial, antifertility, anti-implantation, antimalarial, and anticancer effects [8,13,14,15,16,17,18,19]. As of now, labdane diterpenoids, including leonotin, leonotinin, nepetaefolin, and dubiin, have been isolated from the species, alongside the lanostane triterpenoid 12*β*-acetoxy-20-hydroxy-3,7,11,15-tetraoxo-25,26,27-trisnorlanost-8-en-24-oic acid [20,21]. Leonotin and leonotinin exhibit cytotoxic activity against leukemic cell lines [22]. Notwithstanding its therapeutic importance, *Leonotis ocymifolia* remains inadequately investigated. Therefore, to explore new TNBC drug leads from medicinal plants, in this study, *Leonotis ocymifolia* was investigated and its constituents evaluated for cytotoxic activity against triple-negative breast cancer (HCC70), hormone receptor-positive breast cancer (MCF-7), and non-tumorigenic mammary epithelial cell lines (MCF-12A).

The isolated compounds were assessed computationally for pharmacokinetics and drug likeness utilizing the Swiss ADME online application. This method employs a strong support vector machine (SVM) algorithm utilizing meticulously curated comprehensive datasets of known inhibitors/non-inhibitors and substrates/non-substrates. The ADME/pharmacokinetics analysis sought to evaluate fundamental factors including gastrointestinal absorption, P-glycoprotein-mediated efflux, and the capacity to traverse the blood–brain barrier.

## 2. Materials and Methods

### 2.1. General Experimental Procedure

Column chromatography was carried out using SiO_2_ (Kieselgel-60 GF254, 15 μm, 230–400 mesh Merck, Darmstadt, Germany) on polyamide columns (5 × 60 cm, 200 g) (Germany GmbH). Thin layer chromatography was carried out on precoated silica gel plates (DC Kieselgel 60 F254, Merck, Germany). The plates were visualized using ultraviolet light (Camag, Muttenz, Switzerland) at a short wavelength of 254 nm and a long wavelength of 366 nm. They were further stained with p-anisaldehyde spraying mixture reagent and heated at 100 °C for 2 min to view UV-inactive compounds. Solvents used for column chromatography were of an analytical grade and purchased from Merck and Sigma.

NMR analysis (^1^H and ^13^C NMR) was carried out on a 400 MHz (400.13 MHz; 100.62 MHz) Varian spectrometer at 25 °C. The pure compounds were dissolved in deuterated chloroform (CDCl_3_). Chemical shifts (δ) were expressed in ppm compared to the deuterated solvent. Spin–spin coupling constants (J) were reported in Hertz. Chemical shifts were measured to 7.2 ppm (^1^H) and 77.0 ppm (^13^C) NMR in deuterated chloroform, and structures of isolated compounds were proposed based on spectra interpretation. The Spectrum Two Universal ATR Fourier Transform Infrared (FTIR) spectrophotometer (Spectrum Two, Perkin Elmer, UK) was used to obtain IR spectra. The infrared absorptions were reported in wavenumbers (cm^−1^) expressed in degrees relative to the plane of polarization at the sodium D-line wavelength (λ = 546.3 nm). The melting point was determined using Stuart Melting Point Apparatus SMP10.

A Thermo Scientific Ultimate 3000 Dionex UHPLC system equipped with a QTOF mass spectrometer (Bruker Daltonics Compact, Waltham, MA, USA) was used to establish the molecular ion (*m*/*z*) of the isolated compounds. The UHPLC system consisted of an RS Auto Sampler WPS-3000, Pump HPG-3400 RS, and detector DAD-3000 RS. Separation was achieved using the Acclaim RSLC 120 with C18 column (100 mm × 2.1 mm, i.d., 2.2 µm particle size) kept at 40 °C. The mobile phase comprised of a water–acetonitrile (10:90, *v*/*v*) solvent system, each containing 0.1% formic acid at a flow rate of 0.2 mL/min. The samples were run under isocratic conditions for 5 min. The mass spectrometry was conducted in both negative and positive ion electrospray modes.

### 2.2. Plant Material

Aerial plant material of *Leonotis ocymifolia* was collected from Centurion, Gauteng Province, in South Africa (S 25°52.580′ E 28°06.290′). The plant species was identified at the South African Botanical Institute in Pretoria, South Africa, and assigned a voucher specimen number (7264000).

### 2.3. Extraction and Isolation

The dried, ground leaves (472.71 g) of *L. ocymifolia* were extracted with dichloromethane (DCM) at room temperature using a shaker. The dried crude extract (132.49 g) was subjected to silica gel column chromatography eluted with a hexane/ethyl acetate (EtOAc) solvent system with increasing polarity to give ten fractions (**A**–**J**). Fraction **I** was further fractionated using a 60:40 hexane/EtOAc solvent mixture to give eight subfractions (**IA**–**IG**). Purification of subfraction **IA** (2.53 g) on small CC using 70:30 hexane/EtOAc afforded leonotin (**5**) (10 mg). Further purification of subfraction **IC** (2.94 g) using 70:30 hexane/EtOAc afforded dubiin (**3**) (22.20 mg) and leonotinin (**6**) (182.80 mg). Fractionation of fraction **J** using a 60:40 hexane/EtOAc solvent mixture gave four subfractions (**JA**–**JD**). Purification of subfraction **JC** (1.75 g) using a small column eluted with 70:30 chloroform (CHCl_3_)/EtOAc resulted in the isolation of 13*S* nepetaefolin (**1**) (91.60 mg). Purification of subfraction **JD** using a small column chromatography eluted using 70:30 EtOAc/CHCl_3_ gave nepetaefolin (**2**) (255.20 mg) and nepetaefuran (**4**) (99.40 mg).

### 2.4. Physico-Chemical Properties of Isolated Compounds 1-6

13*S* nepetaefolin (**1**): colorless crystals, mp: 256–257.0 °C, [α]_D_ +76.0° (CHCl_3_, c = 2.3), HR-ESI-MS at *m*/*z* 404.18, calculated for C_22_H_28_O_7,_ 405.1906 [M+H]^+^, IR; 1738, 1715, 1614, 1237, and 1147 cm^−1^. ^1^H NMR (400 MHz, CDCl_3_): δ 6.48 (1H, d, *J* = 2.7 Hz, H-15), 5.18 (1H, bt, *J* = 3.4 Hz, H-6), 5.10 (1H, d, *J* = 11.7 Hz, H-20β), 4.88 (1H, d, *J* = 2.7 Hz, H-14), 4.43 (1H, d, *J* = 10.6 Hz, H-16β), 4.03 (1H, d, *J* = 10.6 Hz, H-16α), 3.97 (1H, d, *J* = 11.7 Hz, H-20α), 2.67 (1H, d, *J* = 3.9 Hz, H-17β), 2.60 (1H, dd, *J* = 15.4, 3.4 Hz, H-7α), 2.37 (1H, d, *J* = 3.9 Hz, H-17α), 2.18 (1H, m, H-12β), 2.01 (1H, bd, *J* = 3.5 Hz, H-5), 1.98 (3H, s, H-22), 1.95 (1H, m, H-12α), 1.84 (1H, m, H-3β), 1.80 (1H, m, H-1α), 1.80 (1H, m, H-2α), 1.80 (1H, m, H-2β), 1.76 (1H, m, H-11β), 1.73 (1H, m, H-1β), 1.60 (1H, dd, *J* = 15.4, 3.4 Hz, H-7β), 1.54 (1H, m, H-3α), 1.45 (1H, ddd, *J* = 14.0, 9.6, 4.9 Hz, H-11α), 1.13 (3H, s, H-18). ^13^C NMR (101 MHz, CDCl_3_): δ 80.15 (C-16), 73.96 (C-20), 68.06 (C-6), 56.51 (C-8), 47.58 (C-17), 47.06 (C-5), 41.09 (C-10), 40.84 (C-4), 39.75 (C-3), 37.54 (C-12), 33.54 (C-1), 32.49 (C-7), 23.69 (C-11), 22.27 (C-18), 20.38 (C-2), 20.38 (C-22) (Table 1).

Nepetaefolin (**2**): colorless crystals, mp: 258.3–260.2 °C, [α]_D_ -11.2° (CHCl_3_, c = 2.5), HR-ESI-MS at *m*/*z* 404.18, calculated for C_22_H_28_O_7,_ 403.1753 [M-H]^−^, IR; 1738, 1719, 1610, 1216, 1143 cm^−1^. ^1^H NMR (400 MHz, CDCl_3_): δ 6.52 (1H, d, *J* = 2.6 Hz, H-15), 5.16 (1H, bt, *J* = 3.3 Hz, H-6), 5.03 (1H, d, *J* = 2.5 Hz, H-14), 5.03 (1H, d, *J* = 11.6 Hz, H-20β), 4.20 (1H, d, *J* = 10.41 Hz, H-16β), 3.95 (1H, d, *J* = 10.4 Hz, H-16α), 3.93 (1H, d, *J* = 11.6 Hz, H-20α), 2.63 (1H, dd, *J* = 15.2, 3.4 Hz, H-7α), 2.56 (1H, d, *J* = 3.9 Hz, H-17β), 2.34 (1H, d, *J* = 3.9 Hz, H-17α), 2.16 (1H, m, H-12β), 2.00 (1H, bd, *J* = 3.5 Hz, H-5), 2.00 (3H, s, H-22), 1.94 (1H, m, H-12α), 1.81 (1H, m, H-3β), 1.79 (1H, m, H-11β), 1.76 (1H, m, H-1α), 1.76 (1H, m, H-2α), 1.75 (1H, m, H-2β), 1.72 (1H, m, H-1β), 1.61 (1H, dd, *J* = 15.2, 3.4 Hz, H-7β), 1.50 (1H, m, H-3α), 1.44 (1H, m, H-11α), 1.10 (3H, s, H-18). ^13^C NMR (101 MHz, CDCl_3_): δ 175.79 (C-19), 170.26 (C-21), 149.37 (C-15), 105.44 (C-14), 92.71 (C-13),85.95 (C-9), 81.06 (C-16), 73.77 (C-20), 67.87 (C-6), 56.07 (C-8), 47.12 (C-17), 46.89 (C-5), 40.99 (C-10), 40.99 (C-4), 39.59 (C-3), 37.46 (C-12), 33.24 (C-1), 32.45 (C-7), 23.50 (C-11), 22.17 (C-18), 20.90 (C-22), 20.31 (C-2) (Appendix A).

Dubiin (**3**): colorless crystals, mp: 185.4–187.2 °C, [α]_D_ -17.2° (CHCl_3_, c = 1.5), HR-ESI-MS at *m*/*z* 390.20, calculated for C_22_H_30_O_6,_ 390.2100 [M+H]^+^, IR; 3500, 3022, 2930, 1760, 1216 cm^−1^. ^1^H NMR (400 MHz, CDCl_3_): δ 7.35 (1H, t, *J* = 1.70 Hz, H-15) 7.22 (1H, s, H-16) 6.25 (1H, bs, H-14) 5.15 (1H, bd, *J* = 3.3 Hz, H-6), 4.68 (1H, d, *J* = 11.2 Hz, H-20β) 4.26 (1H, d, *J* = 11.2 Hz, H-20α) 2.44 (1H, tt, *J* = 20.2, 10.0 Hz, H-12α) 2.44 (1H, tt, *J* = 20.2, 10.0 Hz, H-12β), 2.12 (1H, dt, *J* = 12.1, 5.4 Hz, H-8) 2.06 (1H, m, H-5) 2.02 (3H, s, H-22) 1.89 (1H, m, H-11α), 1.88 (1H, m, H-1α) 1.83 (1H, m, H-3β) 1.78 (1H, m, H-1β) 1.77 (1H, m, H-2α) 1.77 (1H, m, H-2β) 1.75 (1H, m, H-7α), 1.74 (1H, m, H-11β) 1.71 (1H, m, H-7β) 1.50 (1H, m, H-3α) 1.25 (3H, s, H-18) 0.97 (3H, d, *J* = 5.7 Hz, H-17). ^13^C NMR (101 MHz, CDCl_3_): δ 170.19 (C-21),143.23 (C-15), 138.54 (C-16), 124.38 (C-13), 110.49 (C-14), 75.82 (C-20), 74.90 (C-9), 68.33 (C-6), 46.79 (C-5), 41.08 (C-4), 40.96 (C-10), 39.71 (C-3), 35.04 (C-11), 33.32 (C-7), 32.86 (C-1), 30.21 (C-8), 22.36 (C-18), 21.00 (C-22), 20.50 (C-12), 20.38 (C-2), 15.60 (C-17) (Appendix A).

Nepetaefuran (**4**): colorless crystals, mp: 233.6–236.2 °C, [α]_D_ +86.5° (CHCl_3_, c = 3.38), HR-ESI-MS at *m*/*z* 404.1913, calculated for C_22_H_28_O_7,_ 405.1908 [M+H]^+^, IR; 3501, 1726, 1237, and 1143 cm^−1^. ^1^H NMR (400 MHz, CDCl_3_): δ 7.36 (1H, dd, *J* = 1.8 Hz, H-15), 7.22 (1H, bs, H-16), 6.24 (1H, m, H-14), 5.16 (1H, dt, *J* = 3.2 Hz, H-6), 5.01 (1H, d, *J* = 11.8 Hz, H-20β), 4.03 (1H, d, *J* = 11.7 Hz, H-20α), 2.68 (1H, d, *J* = 3.8 Hz, H-17β), 2.64 (1H, d, *J* = 3.2 Hz, H-7α), 2.53 (1H, m, H-12β), 2.34 (1H, m, H-12α), 2.33 (1H, d, *J* = 3.8 Hz, H-17α), 2.11 (1H, d, *J* = 3.3 Hz, H-5), 1.97 (3H, s, H-22), 1.86 (1H, m, H-1α), 1.82 (1H, m, H-3β), 1.76 (2H, m, H-2), 1.74 (1H, m, H-1β), 1.52 (1H, dd, *J* = 15.6, 2.9 Hz, H-7β), 1.50 (1H, m, H-3α), 1.43 (1H, ddd, *J* = 14.9, 9.8, 4.9 Hz, H-11α), 1.11 (3H, s, H-18). ^13^C NMR (101 MHz, CDCl_3_): δ 176.30 (C-19), 170.55 (C-21), 143.56 (C-15), 138.81 (C-16), 124.24 (C-13), 110.66 (C-14), 74.51 (C-9), 73.63 (C-20), 68.06 (C-6), 56.75 (C-8), 46.96 (C-17), 46.45 (C-5), 42.08 (C-4), 41.19 (C-10), 39.96 (C-3), 32.52 (C-1), 32.34 (C-7), 25.88 (C-11), 22.47 (C-18), 21.08 (C-22), 20.48 (C-2), 20.48 (C-12) (Appendix A).

Leonotin (**5**): colorless crystals, mp: 174.5–175.2 °C, [α]_D_ +70.6° (CHCl_3_, c = 1.2), HR-ESI-MS at *m*/*z* 348.19, calculated for C_20_H_28_O_5,_ 349.2001 [M+H]^+^, IR; 3490, 1734, 1264, and 1147 cm^−1^. ^1^H NMR (400 MHz, CDCl_3_): δ 7.34 (1H, t, *J* = 1.6 Hz, H-15), 7.24 (1H, brs, H-16), 6.30 (1H, bs, H-14), 4.78 (1H, dt, *J* = 8.8, 6.2 Hz, H-6), 2.77 (1H, ddd, *J* = 15.4, 10.9, 4.7 Hz, H-12β), 2.59 (1H, ddd, *J* = 15.0, 10.5, 5.8 Hz, H-12α), 2.34 (1H, m, H-7α), 2.34 (1H, m, H-7β), 2.29 (1H, d, *J* = 6.1 Hz, H-5), 2.25 (1H, m, H-11β), 2.12 (1H, m, H-3β), 1.76 (1H, m, H-11α), 1.63 (1H, m, H-2α), 1.51 (1H, m, H-2β), 1.50 (1H, m, H-1α), 1.39 (1H, m, H-1β), 1.36 (3H, s, H-17), 1.35 (1H, m, H-3α), 1.27 (3H, s, H-18), 1.05 (3H, s, H-20). ^13^C NMR (101 MHz, CDCl_3_): δ 182.85 (C-19), 142.97 (C-15), 138.60 (C-16), 125.47 (C-13), 111.01 (C-14), 79.64 (C-9), 75.47 (C-8), 74.70 (C-6), 43.86 (C-5), 42.34 (C-4), 42.01 (C-7), 40.81 (C-10), 32.10 (C-11), 31.33 (C-1), 30.79 (C-17), 28.72 (C-3), 25.42 (C-18), 21.00 (C-20), 20.69 (C-12), 17.84 (C-2) (Appendix A).

Leonotinin (**6**): colorless crystals, mp: 183.3–184.7 °C, [α]_D_ +12.4° (CHCl_3_, c = 1.0), HR-ESI-MS at *m*/*z* 346.18, calculated for C_20_H_28_O_5,_ 347.1855 [M+H]^+^, IR; 3435, 2926, 2858, 1733, 1247, and 1201 cm^−1^. ^1^H NMR (400 MHz, CDCl_3_): δ 7.35 (1H, d, *J* = 1.9 Hz, H-15), 7.22 (1H, bs, H-16), 6.26 (1H, bs, H-14), 4.85 (1H, dt, *J* = 7.5, 5.0 Hz, H-6), 2.93 (1H, d, *J* = 4.7 Hz, H-17β), 2.50 (2H, t, *J* = 8.0 Hz, H-12), 2.44 (1H, d, *J* = 4.6 Hz, H-17α), 2.33 (1H, d, *J* = 4.9 Hz, H-5), 2.30 (2H, bd, *J* = 5.23 Hz, H-7), 2.13 (1H, m, H-3β), 1.76 (2H, bd, *J* = 8.3 Hz, H-11), 1.67 (1H, m, H-2β), 1.52 (1H, m, H-1α), 1.51 (1H, m, H-2α), 1.37 (1H, m, H-3α), 1.33 (1H, m, H-1β), 1.29 (3H, s, H-18), 1.08 (3H, s, H-20). ^13^C NMR (101 MHz, CDCl_3_): δ 182.60 (C-19), 143.14 (C-15), 138.65 (C-16), 124.91 (C-13), 110.73 (C-14), 76.10 (C-9), 74.39 (C-6), 59.31 (C-8), 50.46 (C-17), 45.11 (C-5), 42.82 (C-4), 40.65 (C-10), 32.96 (C-7), 29.91 (C-11), 29.70 (C-1), 28.65 (C-3), 24.38 (C-18), 20.82 (C-20), 20.23 (C-12), 18.08 (C-2) (Appendix A).

### 2.5. Cytotoxic Assay

The cytotoxicity of the isolated compounds was determined by measuring the cell viability of three breast cell lines, namely triple-negative breast cancer HCC70 (ATCC: CRL-2315), hormone receptor-positive breast cancer MCF-7 (ATCC: HTB-22), and non-tumorigenic mammary epithelial cell lines MCF-12A (ATCC: CRL-10,782), upon treatment with the isolated compounds using the MTT assay [23]. The cells were seeded at 5000 cells/well in a 96-well plate and allowed to adhere overnight at 37 °C in a humidified 9% CO_2_ incubator. The cells were treated with DMSO as a vehicle control, 15 nM of paclitaxel (Sigma-Aldrich) as a positive control, and a six-point range concentration (0.32 to 200 µM) of the isolated pure compounds and incubated for 96 h at 37 °C in a humidified 9% CO_2_ incubator. Thereafter, an MTT solution (10 µL of 2.5 mg/mL, Sigma-Aldrich) was added to each well and incubated for 4 h, followed by the addition of 10% SDS in 0.01 M HCl overnight. The formazan product was measured at 570 nm using a Powerwave spectrophotometer (BioTek) with a reference wavelength of 630 to 690 nm. The experiment was performed in technical triplicate, and the data were analyzed using GraphPad Prism Software version 4 (GraphPad Inc., Boston, MA, USA). The concentration that inhibits the proliferation of 50% of the cells (half maximal inhibitory concentration, IC_50_ values) was established by non-linear regression from the dose–response curves using Graph Pad Prism [23].

### 2.6. Physiochemical Properties (ADME)

The structure file generator, which is available for free on the SwissADME website, generated the SMILES for each structure. We used the web tool to calculate a number of simple molecular and physicochemical descriptors, including molecular weight (MW), molecular refractivity (MR), count of specific atom types, and topological polar surface area (TPSA), the latter of which has been shown to be a useful descriptor in many models for estimating membrane diffusion, ADME, and pharmacokinetic behavior. The lipophilicity was assessed using five alternative predictive models (XLOGP, WLOGP, MLOGP, SILICOS-IT, and iLOGP) as well as a consensus logP estimation based on the average value of the various computational parameters [24]. Similarly, the aqueous solubility was determined using three different models [25]. The drug similarity analysis was carried out using the validated rules used as high-throughput screen filters in some of the following leading pharmaceutical companies: Lipinski (Pfizer), Ghose (Amgen), Veber (GSK), Egan (Pharmacia), and Muegge (Bayer). The Abbott bioavailability score was calculated to predict the likelihood of a 10% oral bioavailability. These filters were created to assess drug likeness or to predict whether a chemical entity is likely to have useful pharmacokinetic properties by using calculations based on parameters such as molecular weight, logP, number of HPA, and HBD [26]. Furthermore, the feasibility of using the presented structures as starting scaffolds or lead compounds in a future synthetic drug discovery program was assessed using specific medicinal chemistry and lead-likeness filters [25].

## 3. Results and Discussions

### 3.1. Isolated Compounds

Phytochemical investigation of the DCM leaf extract of *L. ocymifolia* resulted in the isolation of bis-spirolabdane 13S-nepetaefolin (**1**) and the furan labdane nepetaefuran (**4**), which was isolated from the species for the first time, together with four labdane diterpenoids, namely nepetaefolin (**2**), dubiin (**3**), leonotin (**5**), and leonotinin (**6**), previously isolated from the species (Figure 2).

13*S*-nepetaefolin (**1**) was obtained as colorless crystals with a specific optical rotation of [α]_D_ +76.0°. The HRMS spectrum gave a molecular ion [M+H]^+^ at *m*/*z* of 405.1906, consistent with the molecular formula C_22_H_29_O_7_. The IR spectrum gave sharp absorption bands characteristic of an ester at 1738 (-COO stretch), lactone at 1715.4 (-COO stretch), and C-O stretch at 1614, 1237, and 1147 cm^−1^. The melting point was found to be in the range of 256.1–257.0 °C.

The ^13^C NMR spectrum showed the presence of twenty-two (22) carbon resonances, which included two methyl carbon resonances at δ 22.27 (C-18) and 21.00 (C-22); nine methylene carbon resonances at δ 33.54 (C-1), 20.38 (C-2), 39.75 (C-3), 32.49 (C-7), 23.69 (C-11), 37.54 (C-12), 80.15 (C-16), 47.58 (C-17), and 73.96 (C-20); four methine resonances at δ 47.06 (C-5), 68.06 (C-6), 107.10 (C-14), and 149.53 (C-15); five quaternary carbon resonances at δ 40.84 (C-4), 56.51 (C-8), 86.01 (C-9), 41.09 (C-10), and 92.41 (C-13); and two carbonyl carbons at δ 175.97 (C-19) and 170.42 (C-21). The presence of the methyl carbon at δ 22.27 (C-18); the six methylene carbons at δ 33.54 (C-1), 20.38 (C-2), 39.75 (C-3), 32.49 (C-7), 23.69 (C-11), and 37.54 (C-12); and the methine at δ 47.06 (C-5) were consistent with a labdane diterpenoid.

The carbonyl carbon resonance at δ 175.97 (C-19) showed HMBC correlation with two downfield methylene proton resonances at δ 5.10 (1H, d, *J* = 11.7 Hz) and δ 3.97 (1H, d, *J* = 11.7 Hz) assigned to H-20α and H-20β, respectively. These resonances were seen corresponding to an alkoxy methylene carbon resonance at δ 73.96 (C-20) in the HSQC spectrum. The H-20α protons were further seen correlating with two quaternary carbon resonances at δ 86.01 (C-9) and 41.09 (C-10), the methine carbon resonance at δ 47.06 (C-5), and a methylene carbon resonance at δ 33.54 (C-1) in the HMBC spectrum. This confirmed the position of the δ lactone ring between C-19 and C-20. Furthermore, the C-1 carbon was seen corresponding with the methylene proton resonances at δ 1.80 and 1.73 in the HSQC spectrum, which were assigned H-1α and H-1β, respectively. The H-1α proton showed coupling with the H-2β proton resonance at δ 1.80 and in turn with the H-3α proton resonance at δ 1.54 in the COSY spectrum.

More correlations were observed; the presence of a carbonyl carbon resonance at δ 170.42 (C-21) and a methyl carbon at δ 20.38 (C-22), typical of an ester group, were observed in the ^13^C NMR spectrum. The previously assigned C-22 methyl carbon was seen corresponding to the methyl proton resonance at δ 1.98 (3H, s) assigned to 3H-22 in the HSQC spectrum. Meanwhile, a correlation between the C-21 carbonyl carbon and the methyl protons at δ 1.98 (3H-22) was observed in the HMBC spectrum. The observed correlation, coupled with the strong carbonyl stretch at 1738 cm^−1^ in the IR spectrum, corroborated the presence of the ester group.

Furthermore, the 3H-22 protons were observed correlating with the methine carbon at δ 68.06 in the HMBC spectrum placed in position (C-6). The C-6 carbon resonance was seen corresponding with a downfield proton resonance at δ 5.18 (1H, bd, *J* = 3.4 Hz) assigned H-6 in the HSQC spectrum. This positioned the ester group on the C-6 carbon in the labdane skeleton. Furthermore, the methine proton at H-6 was seen coupling to the methine proton resonance at δ 2.01 (1H, bd, *J* = 3.6 Hz) assigned to H-5 and the methylene proton resonances at δ 2.63 and 1.59 ascribed to H-7α and H-7β, respectively, in the COSY spectrum. The H-7 proton resonances showed further correlation with two quaternary carbons at δ 86.01 (C-9) and 56.51 (C-8) and a methylene carbon at δ 47.58 (C-17) in the HMBC spectrum.

Moreover, the downfield carbon resonance at δ 107.10 (C-14) showed the HMBC correlation with the methine proton resonance at δ 6.48 (1H, d, *J* = 2.7 Hz) assigned to H-15 and with a methylene proton resonance at δ 4.03 (1H, d, *J* = 10.6 Hz) assigned to H-16. The H-15 proton resonance was seen coupling with the H-14 proton resonances at δ 4.88 (1H, d, *J* = 2.7 Hz) in the COSY spectrum, confirming the assignment of the dihydrofuran ring. Additionally, the H-15 methine proton resonance showed HMBC correlation with carbon resonances at δ 92.46 (C-13) and 80.15 (C-16); in turn, C-14 (δ 107.10) showed HMBC correlation with the methylene proton resonance at δ 1.95 and 2.18 ascribed to H-12α and H-12*β*. This corroborated the presence of a spiro dihydrofuran moiety.

The relative stereochemistry of compound **1** was determined using the NOESY experiment. The NOESY spectrum indicated an absence of NOE correlations between H-20 and H-5 (Figure 3); this supported the trans configuration between C-5 and C-10. Furthermore, NOE correlations between H-6/H7α, H6/3H-18, and H6/H5 were observed. This suggested that these protons are co-facial and α-oriented. Meanwhile, NOE correlations between H20*β*/H1*β* and H20*β*/H11*β* were observed; this indicated that they are co-facial and have a *β*-orientation. A NOESY correlation between H14/H17*β* and H16*β*/H1 suggested an *S*-relative stereochemistry at C-13 opposite to that of the known compound nepetaefolin, which has a 13*R* configuration. Thus, the structure of compound 1 at C-13 was assigned tentatively as *S* and trivially named 13*S*-nepetaefolin.

Epimeric C-13 *bis*-spiro prefuran labdane diterpenoids are a common occurrence in the genus *Leonotis*; this includes 13*S* and 13*R* premarrubiin previously isolated from *Leonotis leonurus* [27,28]. This study reports the first isolation of the 13*S* nepetaefolin epimer in the genus *Leonotis* and in the *Lamiaceae* family.

Nepetaefolin (**2**) was obtained as colorless crystals with a melting point range of 258.3–260.2 °C comparable to the literature melting point of 260 °C [28]. The TOF-MS showed a molecular ion [M-H]^−^ peak at *m*/*z* 403.1753 consistent with the molecular formula C_22_H_28_O_7_. The IR spectrum showed absorptions at 1738 cm^−1^ (C=O) and 1719.2 cm^−1^ (C=O) and 1610, 1216, and 1143 cm^−1^ (C-O). Analysis of the ^1^H NMR and ^13^C NMR spectra (Appendix A) showed that compound **2** is similar to compound **1**. However, it was observed that the carbon resonances at C-14 and C-16 characteristic of the spirofuran moiety were slightly upfield (C-14) and downfield (C-16) in comparison to those of compound **1**. The difference in NMR resonances suggested that the compounds were epimers at C-13. The experimental data for compound **2** compared well with literature values, and thus, this was identified as nepetaefolin, previously isolated from *L. ocymifolia* as well as *L. nepetaefolia* and *L. leonurus* [21,29,30].

Dubiin (**3**) was obtained as colorless crystals. The TOF-MS showed a molecular mass of *m*/*z* of 390.2100 [M+H]^+^ consistent with a molecular formula C_22_H_30_O_6_. The IR showed absorptions at 3500 cm^−1^ (O-H), 1760 cm^−1^ (C=O), 3022 cm^−1^ and 2930 cm^−1^ (C-H), and 1216 cm^−1^ (C-O). A melting point range of 185.4–187.2 °C was obtained comparable to the literature melting point range of 187–188 °C [30]. The ^1^H NMR and ^13^C NMR spectra (Appendix A) of compound **3** were observed to be similar to that of compound **2** except for the absence of the bis-spiro ether and the presence of a 3-(furan-3-yl)-propanol. The ^1^H NMR spectrum showed the presence of three olefinic methine proton resonances at δ 6.25, 7.22, and 7.35 ppm, indicative of a monosubstituted furan. Further 1D and 2D NMR analysis and literature comparisons identified compound **3** as dubiin, previously isolated from *L. ocymifolia* [31].

Nepetaefuran (**4**) was obtained as a colorless crystals. The TOF MS showed a molecular peak [M+H]^+^ ion at *m*/*z* of 405.1908, which corresponded with the molecular formula C_22_H_28_O_7_. The infrared spectrum showed absorptions at 3501 cm^−1^ (O-H), 1726 cm^−1^ (C=O), and 1237–1143 cm^−1^ (C-O). A melting point range of 232 to 234 °C was obtained, which was comparable to that reported for nepetaefuran (233–235 °C) by [31]. The ^13^C NMR spectrum (Appendix A) showed the presence of twenty-two carbon resonances, which included two methyl carbon resonances at δ 22.47 and δ 21.08, eight methylene carbon resonances (δ 32.52, 20.48, 39.96, 32.34, 25.88, 20.36, 46.96, and 73.63), five methine resonances (δ 46.45, 68.06, 110.66, 143.56, and 138.81 ppm), five quaternary carbon resonances (δ 42.08, 41.19, 56.75, 74.51, and 124.24 ppm), and two carbonyl carbons at δ 176.29 and 170.55 ppm. The presence of the methyl carbon at δ 22.47 (C-18) ppm; the six methylene carbons at δ 32.52 (C-1), 20.48 (C-2), 39.96 (C-3), 32.34 (C-7), 25.88 (C-11), and 20.36 (C-12); and the methine at δ 46.45 ppm (C-5) were indicative of a labdane diterpenoid. The ^1^H NMR and ^13^C NMR spectra (Appendix A) of compound **4** were observed to be similar to that of compound **3** except for the presence of a spiro epoxide. The presence of an alkoxy methylene carbon resonance at δ 46.96 and a quaternary carbon resonance at δ 56.75 ppm, indicative of an exocyclic epoxide ring, were observed in the ^13^C NMR spectra. The structure of compound **4** was elucidated and identified as nepetaefuran based on further 1D and 2D NMR analysis and comparison of experimental data to the literature values [32]. Nepetaefuran is among the predominant labdane diterpenoids in the genus *Leonotis* and *Leonurus*. It has been isolated from the species *Leonotis nepetifolia* [32] and the species *Leonurus sibiricus* [22]. However, according to the author’s best knowledge, this is the first report of this labdane diterpenoid in the species, *Leonotis ocymifolia*.

Leonotin (**5**) was obtained as colorless crystals and gave a melting point range of 174.5–175.2 °C similar to the literature melting point of 175 °C [33]. The electrospray ionization–time of flight (ESI+TOF) mass spectrum showed a molecular ion [M+H]^+^ peak at *m*/*z* 349.2001 corresponding to a molecular formula of C_20_H_28_O_5_. The molecular formula was consistent with a double-bond equivalence of seven. The IR spectrum showed absorptions at 3490.4 cm^−1^ (O-H), 1734.2 cm^−1^ (C=O), and 1264.2 and 1147.6 cm^−1^ (symmetrical and asymmetrical C-O). The ^1^H NMR and ^13^C NMR (Appendix A) displayed peaks characteristic of a furan labdane diterpenoid with carbonyl resonance at δ 182.96 and an alkoxy carbon resonance at δ 74.81 ppm indicative of a gamma lactone ring. Comparison of NMR data with the literature values identified compound **5** as leonotin, previously isolated from *L. ocymifolia* [33]. Leonotin has also been isolated from *L. nepetifolia* [33,34] and *Leonurus sibiricus* [22].

Leonotinin (**6**) was obtained as colorless crystals. The electrospray ionization–time of flight (ESI+TOF) mass spectrum showed a molecular ion [M+H]^+^ peak at *m*/*z* 347.1855 corresponding to a molecular formula of C_20_H_26_O_5_. And a double-bond equivalence of seven was deduced. The IR spectrum showed absorptions at 3433 cm^−1^ (O-H) and 1733 cm^−1^ (C=O). A melting point range of 183.3–184.7 °C was obtained comparable to the literature melting point range of 184–186 °C of leonotinin previously obtained by [35]. The ^1^H NMR and ^13^C NMR data (Appendix A) displayed peaks characteristic of a furan labdane diterpenoid similar to compound 5; however, the presence of a spiro epoxide instead of a hydroxyl group was observed. Comparison of the experimental data to that from the literature identified compound **5** as a leonotinin, a derivative of leonotin at carbon 8. Leonotinin was first isolated from the plant *Leonotis nepetifolia* in 1974 by Purushothaman and co-workers. Labdane was recently isolated from the leaves of *Leonotis ocymifolia* by [21], where ^1^H NMR and ^13^C NMR assignments were reassigned.

### 3.2. Cytotoxic Activities

The isolated compounds were evaluated for cytotoxic activity by measuring the half-maximal inhibitory concentrations (IC_50_) of the compounds against triple-negative breast cancer (HCC70), hormone receptor-positive breast cancer (MCF-7), and non-tumorigenic mammary epithelial cell lines (MCF-12A) using the MTT assay. The selectivity index (SI) was assessed for each molecule against the HCC70 cell line, as shown in Table 2. SI function is a crucial metric for assessing a compound’s potential safety, with values below 1 indicating toxicity and those above 10 implying high selectivity for further investigation [36].

The isolated compounds and crude extract showed modest to weak cytotoxicity against the HCC70 cell line (Table 2), with IC_50_ values of 24.65 ± 1.18 µM (13*S*-nepetaefolin (**1**)), 37.76 ± 1.78 µM (DCM crude extract), 73.66 ± 1.10 µM (nepetaefuran (**4**)), 94.89 ± 1.10 µM (leonotinin (**6**)), and 127.90 ± 1.23 µM (dubiin (**3**)). However, they displayed no cytotoxic activity against the MCF-7 cell line (IC_50_ > 200 µM) or non-cancerous MCF-12A cells, except for 13S-nepetaefolin (**1**), which had an IC_50_ of 26.55 ± 1.32 µM against the MCF-12A cell line. The crude extract demonstrated the highest selectivity towards the HCC70 cell line with SI of 5.297 compared to the isolated compounds nepetaefuran (**4**), leonotinin (**6**), dubiin (**3**), and 13S-nepetaefolin (**1**) with SI of 2.715, 2.108, 1.564, and 1.077, respectively.

From these data, it was observed that *bis*-spirolabdane 13*S*-nepetaefolin (**1**) exhibited the highest cytotoxic activity of the extracted compounds, with an IC_50_ value of 24.65 µM against the HCC70 TNBC cell line; however, the compound was equally toxic to non-tumorigenic breast epithelial cells (MCF12A: IC_50_ value of 26.55 µM) with a selectivity index (SI) value of 1.077. Moreover, 13*S*-nepetaefolin (**1**) was not toxic (IC_50_ > 200 μM) to hormone-responsive breast cancer cells (MCF7). It was also observed that its isomer nepetaefolin (**2**) showed no cytotoxic activity (IC_50_ > 200 µM) against any of the cell lines. This suggests that the stereochemistry at C-13 might have an effect on the cytotoxic activity of the *bis*-spirolabdane diterpenoids. The effect of C-13 stereochemistry on the biological activity of labdane diterpenoids was previously reported for the glycoside labdane diterpenoids koraienside I and J from *Pinus koraiensis* [37]. The study found that koraienside **I**, which has a C-13*E* configuration, exhibited better neuroprotective activity than koraienside **J** (C-13Z). Also, from the same study, the labdane diterpenoids koraienside **E** and **F** were found to exhibit different anti-inflammatory activities due to the difference in C-6 hydroxyl configuration.

On the other hand, nepetaefuran (**4**) and leonotinin (**6**) exhibited modest cytotoxic activity against HCC70 TNBC cells, with IC_50_ values of 73.66 and 94.89 µM, respectively. Moreover, these labdanes showed no cytotoxic activity towards the non-tumorigenic breast epithelial cell line (MCF-12A) or hormone-responsive breast cancer cell line (MCF-7). These labdane diterpenoids were previously found to exhibit moderate cytotoxic activity (IC_50_ = 50–60 µg/mL) against the leukemic cell line (L1210) [22]. Furthermore, these were found to possess anti-inflammatory activity by inhibiting the LPS pathway via suppression of the transactivation of NF-kappa B [38]. NF-kappa B is considered a potential molecular drug target in TNBC [39]. Furthermore, dubiin (**3**) exhibited the lowest cytotoxic activity of the extracted compounds against HCC70 cells, with IC_50_ values of 127.90 µM, and it was non-toxic to MCF-12A and MCF-7 cells. The difference in cytotoxic activity between the derivatives dubiin (**3**) and nepetaefuran (**4**) suggests that oxygenation at carbon 8 improves the cytotoxic activity of these furan labdane diterpenoids.

Overall, the crude extract and the isolated compounds demonstrated moderate selective cytotoxicity toward the HCC70 cancer cell line, which suggests that the anticancer activity of *Leonotis ocymifolia* against the TNBC cell line could be attributed to the presence of labdane diterpenoids.

### 3.3. ADME Properties of the Isolated Compounds

Recent studies have shown that labdane exhibits significant cytotoxic and cytostatic effects against leukemic cell lines of human origin and interferes with the biochemical pathways of apoptosis and the cell cycle phases and also functions as an inhibitor of inflammatory cytokines production through NFkB repression [40,41]. Several studies have highlighted labdanes’ anticancer properties, particularly their ability to inhibit the growth and induce the death of glioblastoma and carcinoma cell lines in vitro [42].

In this study, the adsorption, distribution, metabolism, and excretion properties (ADME) of the labdane diterpenoids bis-spirolabdane 13*S*-nepetaefolin (**1**), nepetaefolin (**2**), dubiin (**3**), nepetaefuran (**4**), leonotin (**5**), and leonotinin (**6**) were further predicted using SwissADME tools [43]. This web server was chosen because it is freely available and provides a reliable and fast computational method for estimating the pharmacokinetics and toxicity of small molecules.

The basic physiochemical and pharmacokinetic properties, drug nature, and medicinal chemistry friendliness of the labdane diterpenoids (**1–6**) were studied, and the results are summarized in Table 3.

Based on the findings, all the compounds demonstrated water solubility, with the calculated aqueous solubility descriptor (logP (Ali) ranging from the lowest negative values of 3.10 to 4.00. (Table 3). Based on the calculated logP values, all tested compounds proved to be lipophilic. Conversely, these findings showed that the target compounds are moderately soluble to soluble, depending both on the logS estimation model and the tested compound. The values ranged from −3.99 to 0 in comparison to the Food and Drug Administration’s (FDA) standard drugs [43], Understanding the water solubility properties of these labdane derivatives simplifies the handling and formulation of potential therapeutic agents, facilitating future drug development. In addition, the compounds had positive lipophilicity values (logP (iLOGP) ranging from 2.35 to 2.85). High positive values indicate high lipophilicity. The distribution of logP (iLOGP) values was comparable to most FDA drug values ranging from 0 to 10 but on the lower end [43]. The lipophilicity of a compound is important in drug discovery efforts because it affects its permeability through the biological membrane. Permeability can be reduced if the lipophilicity is too low, and hydrophilic compounds cannot diffuse through the membrane [43,44].

Lipophilicity influences drug ADMET characteristics, including solubility and membrane permeability [45,46]; potency [47,48], selectivity, and promiscuity; metabolism and pharmacokinetics [47]; and pharmacodynamic and toxicological profiles [47,48]. When comparing marketed oral drugs to compounds in the early stages of development, it is common to find that compounds with high lipophilicity (>5) have rapid metabolic turnover [47,48], low solubility, and poor absorption [46,48]. If lipophilicity is excessively high, there is a higher risk of in vitro receptor promiscuity [47], in vivo toxicity, poor solubility, and metabolic clearance. If a drug’s lipophilicity is too low, it will have poor ADMET properties. Lipinski recommended a lipophilicity range of logP < 5 for compounds entering Phase II clinical trials [49]. However, Gleeson suggested that compounds with logP < 4 (and a molecular weight < 400) have a higher chance of success against a broad set of ADMET parameters. A recent literature review suggested that the optimal region of lipophilicity lies within a narrow range of log D between –1 and 3.

Furthermore, the metabolism prediction revealed that leonotin (**5**) and leonotinin (**6**) inhibited CYP2D6 but not CYPC19. However, all of the compounds demonstrated high gastrointestinal absorption (Table 3), indicating good water solubility, which is an extremely desirable feature of a drug candidate given the undeniable benefits of the oral route of administration. Furthermore, leonotin (**5**) and leonotinin (**6**) exhibited lead-like properties and high synthetic accessibility scores when compared to other labdanes, as shown in Table 3. The skin permeation ability of the tested diterpenoids is expected to be very low based on the calculated logKp values. All compounds were found to comply with the Lipinski rules, which include the pioneering drug candidate filter used in Pfizer’s drug discovery screens and are regarded as the ultimate archetype of all drug-similarity tools. In contrast, the tested diterpenoids had no violations of the rules implemented in the Veber filter or Ghose and Egan filters. Because of their chemical complexity, high molecular mass, and lipophilicity, the tested series generally failed to comply with the Muegge and Brent lead-likeness filters [25], indicating that if they are to be used as starting scaffolds for drug discovery programs, the synthetic strategies should be focused on structure simplification, elimination of troublesome functionalities, and decreased lipophilicity.

The bioavailability radar in Figure 4 revealed the optimal range for each property for the majority of the compounds [43]. Leonotin (**5**) and leonotinin (**6**) demonstrated the highest bioavailability, indicating their potential as drug candidates, whereas compounds (**1** and **2**), dubiin (**3**), and nepetaefuran (**4**) exhibited poor flexibility.

## 4. Conclusions

Previously unreported bis-spiro labdane (**1**) and four previously obtained labdane diterpenoids (**2**–**6**) were isolated from the dichloromethane leaf extract of *Leonotis ocymifolia*. The isolated labdane diterpenoids exhibited moderate selective cytotoxicity against triple-negative breast cancer cells (HCC70), with bis-spiro labdane diterpenoid demonstrating the highest cytotoxic activity among all compounds, presenting an IC_50_ of 24.65 μM, albeit with comparable toxicity to the non-cancer control cell line (MCF-12A). The compound labdane 13*S*-nepetaefolin demonstrated cytotoxic effects on HCC70 cells, but its isomer (**2**) exhibited no such effects, suggesting that the stereochemistry at C-13 influences the cytotoxic activity of bis-spirolabdane diterpenoids against the HCC70 breast cancer cell line. All compounds (**1**–**6**) exhibited adsorption, distribution, metabolism, and excretion (ADME) characteristics, while leonotin (**3**) and leonotinin (**6**) displayed lead-like features and elevated synthetic accessibility scores. The preliminary findings from this study necessitate additional exploration of *L. ocymifolia* for potential TNBC therapeutic agents. Future studies should focus on modification of functional groups of bis-spiro labdane diterpenoid at positions that are critical for binding to specific receptors or enzymes overexpressed in breast cancer cells.

## Figures and Tables

**Figure 1 diseases-13-00140-f001:**
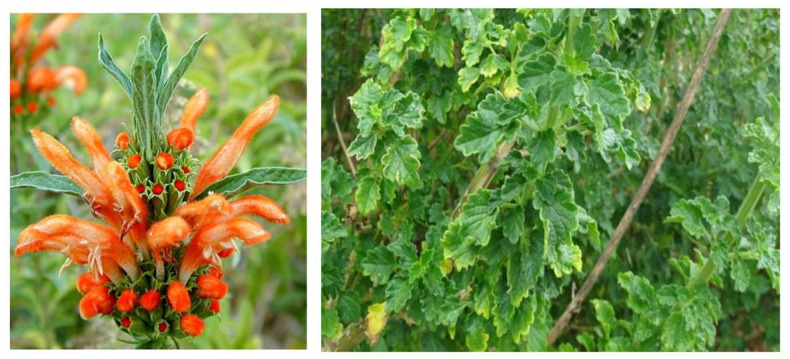
Aerial parts of *L. ocymifolia*.

**Figure 2 diseases-13-00140-f002:**
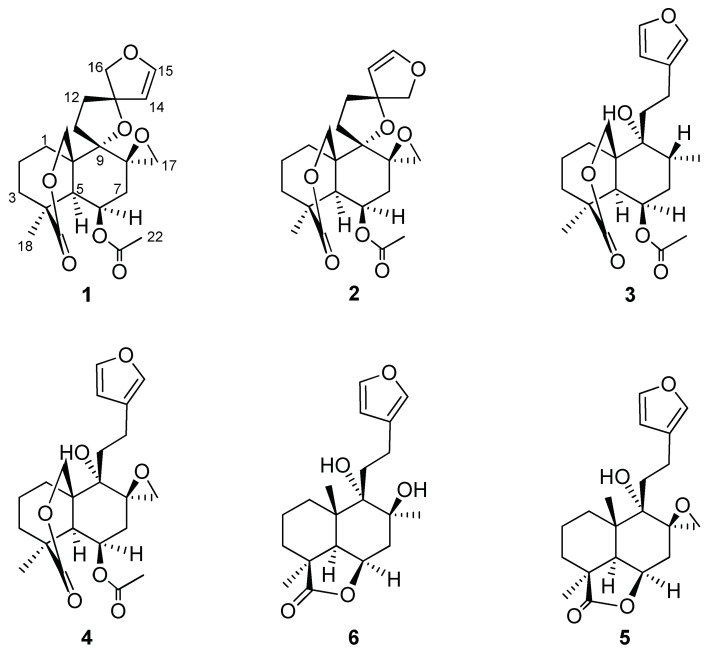
Structures of labdane diterpenoids isolated *L. ocymifolia*.

**Figure 3 diseases-13-00140-f003:**
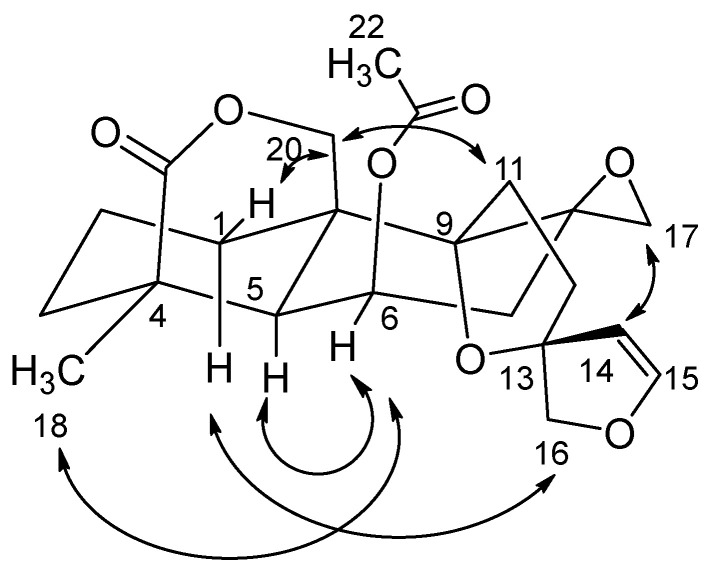
NOE correlations of compound **1**.

**Figure 4 diseases-13-00140-f004:**
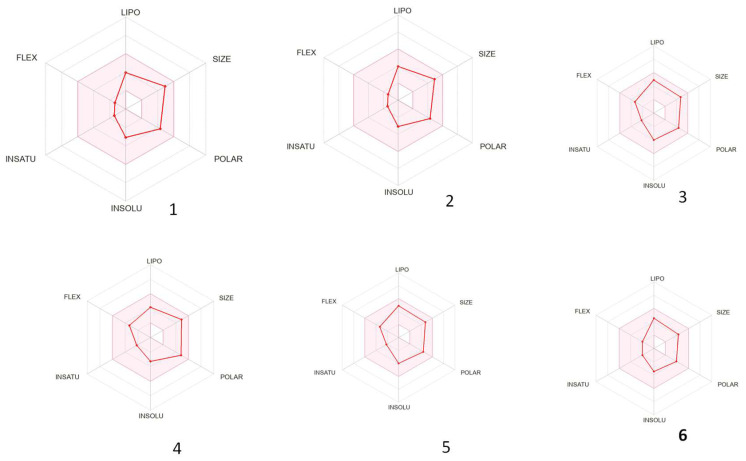
Bioavailability radar of cytotoxic compounds. The pink region indicates the optimal range of the stated properties.

**Table 1 diseases-13-00140-t001:** ^1^H NMR (400 MH_Z_) and ^13^C (100.62 MHz) data for compounds **1** in CDCl_3_.

Compound 1					
Carbon No.	δ ^1^H NMR (J in Hz)	δ ^13^C NMR (J in Hz)	COSY	HMBC	NOESY
1 α	1.80 (1H, m)	33.07 (CH_2_)	H-1 β	C-5	
1 β	1.73 (1H, m)		H-1 α	C-5	
2 α	1.80 (1H, m)	20.32 (CH_2_)	H-2 β	C-10, C-4	
2 β	1.80 (1H, m)		H-2 α	C-10, C-4	
3 α	1.54 (1H, m)	39.69 (CH_2_)	H-3 β	C-4	
3 β	1.84 (1H, m)		H-3 α	C-4	
4	-	40.77 (C)	-	-	-
5	2.01 (1H, d, *J* = 3.6 Hz)	46.98 (CH)	H-6	C-20, C-10, C-19	H-6
6	5.18 (1H, d, *J* = 3.4 Hz)	67.97 (CH)	H-5		H-18, H-7α, H-5
7 α	2.63 (1H, dd, *J* = 15.4, 3.4 Hz)	32.43 (CH_2_)	H-7 β, H-6	C-6, C-8, C-9, C-17	H-6
7 β	1.59 (1H, dd, *J* = 15.4, 3.4 Hz)		H-7α, H-6	C-6, C-8, C-9, C-17	
8	-	56.44 (C)	-	-	-
9	-	85.92 (C)	-	-	-
10	-	41.02 (C)	-	-	-
11 α	1.45 (1H, ddd, *J* = 14.0, 9.6, 4.9 Hz)	23.63 (CH_2_)	H-11 β	C-13, C-9, C-12	
11 β	1.76 (1H, m)		H-11 α	C-13, C-9, C-12	
12 α	1.95 (1H, m)	37.46 (CH_2_)	H-12 β	C-9, C-15, C-13, C-16	
12 β	2.18 (1H, m)		H-12 α	C-9, C-13, C-16	
13	-	92.41 (C)	-	-	-
14	4.88 (1H, d, *J* = 2.7 Hz)	107.01 (CH)	H-15	C16, C13, C15	H-15, H-17β
15	6.48 (1H, d, *J* = 2.7 Hz)	149.43 (CH)	H-14	C13, C16	H-14
16 α	4.03 (1H, d, *J* = 10.6 Hz)	80.07 (CH_2_)	H-16 β	C13, C12	
16 β	4.43 (1H, d, *J* = 10.6 Hz)		H-16 α	C15, C13	
17 α	2.34 (1H, d, *J* = 3.9 Hz)	47.53 (CH_2_)	H-17 β	C-8	
17 β	2.67 (1H, d, *J* = 3.9 Hz)		H-17 α	C-8	H-14
18	1.07 (3H, s)	22.21 (CH_3_)	-	C-19, C-3, C5, C4	H-6
19	-	175.91 (C)	-	-	-
20 α	3.97 (1H, d, *J* = 11.7 Hz)	73.9 (CH_2_)	H-20 β	C1, C10, C5, C19	
20 β	5.10 (1H, d, *J* = 11.7 Hz)		H-20 α	C9, C5, C1	
21	-	170.33 (C)	-	-	-
22	1.98 (3H, s)	20.95 (CH_3_)	-	C-21	

**Table 2 diseases-13-00140-t002:** Cytotoxic activities (IC_50_) of labdane diterpenoids against breast cancer cells and non-cancer equivalents.

Compound	HCC70 (IC_50_ and SD) R^2^	MCF-7 (IC_50_ and SD) R^2^	MCF-12A (IC_50_ and SD) R^2^	Selectivity Index (SI) MCF-7	Selectivity Index (SI) HCC70
13S-Nepetaefolin (**1**)	24.65 ± 1.18 0.9656	NT	26.55 ± 1.32 0.9968	<0.132	1.08
13R-Nepetaefolin (**2**)	NT	NT	NT	N/A	N/A
Dubiin (**3**)	127.90 ± 1.23 0.8173	NT	NT	N/A	1.56
Nepetaefuran (**4**)	73.66 ± 1.10 0.9689	NT	NT	N/A	2.72
Leonotinin (**6**)	94.89 ± 1.10 0.9417	NT	NT	N/A	2.11
Leonotis DCM crude extract	37.76 ± 1.78 0.9401	NT	NT	N/A	5.30
Paclitaxel (nM)	3.920 ± 1.03 0.9920	2.410 ± 1.11 0.9743	16.16 ± 1.08 0.9877	6.71	4.12

Not toxic = NT > 200 µM.

**Table 3 diseases-13-00140-t003:** The ADME properties of isolated compounds from *L. ocymifolia*.

Analysis	1	2	3	4	5	6
	Water solubility					
LogS (ESOL)	−3.10	−3.10	−4.00	−3.28	−3.53	−3.47
Solubility	3.18 × 10^−1^ mg/mL; 7.87 × 10^−4^ mol/L	3.18 × 10^−1^ mg/mL; 7.87 × 10^−4^ mol/L	2.13 × 10^−1^ mg/mL; 5.26 × 10^−4^ mol/L	3.87 × 10^−2^ mg/mL; 9.92 × 10^−5^ mol/L	1.04 × 10^−1^ mg/mL; 2.98 × 10^−4^ mol/L	1.16 × 10^−1^ mg/mL; 3.35 × 10^−4^ mol/L
Class	Soluble	Soluble	Soluble	Moderately soluble	Soluble	Soluble
LogS (Ali)	−2.77	−2.77	−4.55	−3.49	−3.82	−3.60
Solubility	1.66 × 10^−1^ mg/mL; 4.11 × 10^−4^ mol/L	1.66 × 10^−1^ mg/mL; 4.11 × 10^−4^ mol/L	8.26 × 10^−3^ mg/mL; 2.04 × 10^−5^ mol/L	5.43 × 10^−3^ mg/mL; 1.39 × 10^−5^ mol/L	5.23 × 10^−2^ mg/mL; 1.50 × 10^−4^ mol/L	8.71 × 10^−2^ mg/mL; 2.52 × 10^−4^ mol/L
Class	Soluble	Soluble	Moderately soluble	Moderately soluble	Moderately soluble	Soluble
Log*S* (SILICOS-IT)	−3.39	−3.39	−4.69	−4.86	−4.57	−4.77
Solubility	1.66 × 10^−1^ mg/mL; 4.11 × 10^−4^ mol/L	1.66 × 10^−1^ mg/mL; 4.11 × 10^−4^ mol/L	8.26 × 10^−3^ mg/mL; 2.04 × 10^−5^ mol/L	5.43 × 10^−3^ mg/mL; 1.39 × 10^−5^ mol/L	9.49 × 10^−3^ mg/mL; 2.72 × 10^−5^ mol/L	5.94 × 10^−3^ mg/mL; 1.71 × 10^−5^ mol/L
Class	Soluble	Soluble	Moderately soluble	Moderately soluble	Moderately soluble	Moderately soluble
	Physiochemical properties					
No. of heavy atoms	29	29	28	29	25	25
No. of aromatic heavy atoms	0	0	5	5	5	5
No. of rotatable bonds	2	2	5	5	3	3
No. of H-bonds acceptors	7	7	6	7	5	5
No. of H-bonds donors	0	0	1	1	2	2
Molar refractivity	100.13	100.13	102.30	101.31	96.60	72.20
Gastrointestinal absorption	High	High	High	High	High	High
CYPC19 inhibitor	No	No	No	No	No	No
CYP1A2 inhibitor	No	No	No	No	No	No
CYP2C9 inhibitor	No	No	No	No	No	No
CYP3A4 inhibitor	No	No	Yes	Yes	No	No
CYP2D6 inhibitor	No	No	No	Yes	Yes	Yes
Log K_p_ (skin penetration) in cm/s	−7.77	−7.77	−6.50	−7.49	−2.88	−2.73
P-gp substrate	No	No	No	No	Yes	Yes
	Drug likeness					
Lipinski	Yes; violation	Yes; 0 violation	Yes; 0 violation	Yes; 0 violation	Yes; 0 violation	Yes; 0 violation
Ghose	Yes	Yes	Yes	Yes	Yes	Yes
Veber	Yes	Yes	Yes	Yes	Yes	Yes
Egan	Yes	Yes	Yes	Yes	Yes	Yes
Muegge	Yes	Yes	Yes	Yes	Yes	Yes
Bioavailability score	0.55	0.55	0.55	0.55	0.55	0.55
	Medicinal chemistry					
Lead likeness	No; Violation; MV < 350	No; Violation; MV < 350	No; Violation; MV < 350	No; Violation; MV < 350	Yes	Yes
Synthetic accessibility	6.40	6.40	5.61	5.81	4.95	5.29
	Lipophilicity					
Implicit logP (iLOGP)	2.35	2.35	2.72	2.57	2.85	2.73
Log*P*_o/w_ (XLOGP3)	1.41	1.41	1.80	3.08	2.50	2.44
Log*P*_o/w_ (WLOGP)	2.27	2.27	2.40	3.26	2.84	2.85
Log*P*_o/w_ (MLOGP)	1.62	1.62	1.48	2.28	1.93	1.93
Log*P*_o/w_ (SILICOS-IT)	3.02	3.02	3.65	3.68	3.46	3.95
Consensus log*P*_o/w_	2.13	2.13	3.00	2.38	2.72	2.78

## Data Availability

Data are contained within the article and Appendix A.

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
