# Peer review of "Labdane Diterpenoids from Leonotis ocymifolia with Selective Cytotoxic Activity Against HCC70 Breast Cancer Cell Line"

_diseases, 2025, doi:10.3390/diseases13050140_

Round 1
Reviewer 1 Report
Comments and Suggestions for Authors
In the manuscript described structure and anticancer activity of six compound isolated from Leonotis ocymifolia. The chemical structure of compounds was characterized using NMR, MS and IR spectroscopies. The tested compounds show high anticancer activity against triple negative breast cancer cells (HCC70). The in silico ADMET parameters shows that compounds should characterized high bioavailability.
I suggest the Authors to pay more attention to the following comments point by point:
- The NMR data should be added to paragraph 2.4.
- The spectra of all compounds should be added in Supplementary materials.
- The selectivity index (SI) should be added to Table 2
Author Response
I suggest the Authors to pay more attention to the following comments point by point:
- The NMR data should be added to paragraph 2.4.
Response:
Thank you for the constructive comments and suggestions. The NMR data has been added to section 2.4 accordingly and highlighted in yellow.
- The spectra of all compounds should be added in Supplementary materials.
Response:
The spectra of all compounds have been added in Supplementary materials and uploaded together with the manuscript.
- The selectivity index (SI) should be added to Table 2
Response:
The selectivity index (SI) has been calculated and added as in Table 2.
Reviewer 2 Report
Comments and Suggestions for Authors
Include more anticancer studies data like ROS staining, dual staining, and cell cycle arrest
Include compound-treated cancer cell images
What about other biomedical applications? Include more biomedical applications
The journal, like “disease”, authors need to concentrate on disease models
Include the Leonotis ocymifolia shrub images
Rewrite the introduction part
Check the minor errors throughout the manuscript. For example, no space between Cytotoxicactivities
Check grammatical errors throughout the manuscript
Comments on the Quality of English LanguageNeed to improve
Author Response
- Include more anticancer studies data like ROS staining, dual staining, and cell cycle arrest
The journal, like “disease”, authors need to concentrate on disease models
Response
Thank you for your comments and suggestions. Your suggestions are certainly an intriguing avenues for future investigation. However, as it falls outside the current scope, we were unable to include these studies at the time. We believe that these findings could be a valuable part of subsequent research that will builds on our current findings, and we would be happy to explore them further in future work.
- Include compound-treated cancer cell images
Response
Thank you, unfortunately the compound-treated cancer cell images for this study were lost.
- What about other biomedical applications? Include more biomedical applications
Response
Other applications of labdanes diterpenoids have been included and highlighted in the introduction
- Include the Leonotis ocymifoliashrub images
Response:
Image of Leonotis ocymifolia has been added to the article as Figure 1.
- Rewrite the introduction part
Response:
Introduction rewritten
- Check the minor errors throughout the manuscript. For example, no space between Cytotoxicactivities
Response:
Minor errors corrected throughout the manuscript
- Check grammatical errors throughout the manuscript
Response:
We have rechecked the manuscript for typos and grammatical mistakes using Grammarly and the manuscript has been proof read using a native English speaker.
Reviewer 3 Report
Comments and Suggestions for Authors
This is a well-controlled, well-written and well-designed study, demonstrating the selective cytotoxicity of bioactive molecules from Leonotis ocymifolia against TNBC- an important health topic to address.
Please include what the acronyms stand for, the first time they are used- such as TNBC in the abstract.
Trypan blue exclusion assay may be a useful assay to add to the study as this assay will clearly help distinguish between living vs. dead cells that lost their membrane integrity. MTT is a very useful test yet it is more robust in indicating proliferation. Alternatively, please comment on this drawback.
One of the most pressing issues regarding breast cancer is metastasis – can the authors elaborate on the potential effect of the extracts on inhibiting metastasis, by including references? Alternatively, can a wound heal assay be included (using concentrations below MTT cut off) to see if the extracts also display antimetastatic activity?
Addition of biological replicates would increase the reliability of results.
Author Response
- Please include what the acronyms stand for, the first time they are used- such as TNBC in the abstract.
Response:
Acronyms have been properly defined upon their first use.
- Trypan blue exclusion assay may be a useful assay to add to the study as this assay will clearly help distinguish between living vs. dead cells that lost their membrane integrity. MTT is a very useful test yet it is more robust in indicating proliferation. Alternatively, please comment on this drawback.
Response:
While the Trypan blue exclusion assay determines cell viability by distinguishing between live and dead cells based on membrane integrity. Cells must be assessed immediately after staining to avoid changes in cell status, whereas for the MTT assay it is non-invasive and can indirectly assess cell viability and proliferation.
- One of the most pressing issues regarding breast cancer is metastasis – can the authors elaborate on the potential effect of the extracts on inhibiting metastasis, by including references?
Response:
Although we have not conducted specific antimetastatic activity studies on our extracts and pure compounds, existing research on structurally similar labdane diterpenoids from other plant families suggests potential antimetastatic properties. Studies on labdane diterpenoids such as andrographolide, forskolin have demonstrated that these compounds can inhibit key processes involved in metastasis, such as cell migration and invasion.
To support this discussion, the following reference studies on related compounds have demonstrated antimetastatic activity, however, we have not included in our discussion in the manuscript as we did not conduct antimetastatic properties.
Wu, Z.W., Xu, H.W., Dai, G.F., Liu, M.J., Zhu, L.P., Wu, J., Wang, Y.N., Wu, F.J., Zhao, D., Gao, M.F. and Nie, S.S., 2013. Improved inhibitory activities against tumor-cell migration and invasion by 15-benzylidene substitution derivatives of andrographolide. Bioorganic & Medicinal Chemistry Letters, 23(23), pp.6421-6426.
Pearngam, P., Kumkate, S., Okada, S. and Janvilisri, T., 2019. Andrographolide inhibits cholangiocarcinoma cell migration by down-regulation of claudin-1 via the p-38 signaling pathway. Frontiers in Pharmacology, 10, p.827.
Rao, B.V., Swain, S., Siva, B., Priya, T.S., Alli, V.J., Jadav, S.S., Jain, N., Ramalingam, V. and Babu, K.S., 2024. Novel Bis-spiro-labdane type Diterpenes from Leonotis nepetifolia: Isolation, Semi-synthesis, and Evaluation of their Cytotoxic Activities. Journal of Molecular Structure, 1305, p.137728.
- Alternatively, can a wound heal assay be included (using concentrations below MTT cut off) to see if the extracts also display antimetastatic activity? Addition of biological replicates would increase the reliability of result
Unfortunately we were unable to perform a wound healing assay at this stage. However, the project is continuing and this studies will be explored.
Reviewer 4 Report
Comments and Suggestions for Authors
The study summarizes the findings of laboratory research on the isolation and characterization of labdanic diterpenoids from the plant Leonotis ocymifolia. Six compounds were isolated, among them a new diterpenoid, 13S-nepetaefolin, and their cytotoxicity was tested against the triple-negative breast cancer cell line HCC70. Interestingly, 13S-nepetaefolin was the most cytotoxic, but it also demonstrated the same level of toxicity against non-cancerous MCF-12A cells, whereas its isomeric analogue (2) was inactive. The results indicate that stereochemistry may affect the biological activity of these molecules. Besides, the labdanic diterpenoids were evaluated for their pharmacokinetic properties using the SwissADME tool, where the molecules leonotin (5) and leonotinin (6) showed promise as model compounds for further drug discovery.
It is my view that the research is valuable in that it adds to the identification of novel natural compounds possessing anticancer activities against one of the most challenging types of cancer, i.e., triple-negative breast cancer. The fact that the isolated compounds were selectively cytotoxic against HCC70 cells is a promising lead towards the development of novel therapy agents. Further, the assessment of the pharmacokinetic profiles of the compounds gives an important indication of the likelihood of their application as drug candidates.
The article can be published if certain important issues are addressed:
-The authors describe the identification of the compounds by spectroscopic methods like NMR, IR, and MS. The spectra are not provided in the supplementary data, however, which would be helpful in verifying the results of identification. The inclusion of such information is necessary to ensure reproducibility and openness of results.
-The figure 1 of the structures of the compounds isolated is of poor resolution (compound 1) and is hard to read.
- Although the introduction is a good summary of why TNBC is so significant and why it's challenging to treat, the fact about the limited therapeutic options is restated. It would be better to have briefer presentation of this and clearer linkage to why this study is needed.
-The cytotoxicity results are given in a table; however, preparatory analysis in the text lacks structure. Specifically, the relative assessment of the activity of the various compounds can be stated more clearly, highlighting the principal findings.
-Though the ADME properties of the compounds are analyzed by SwissADME, the discussion of the results remains relatively brief. A more in-depth exploration of how solubility and lipophilicity influence the potential pharmacological uses of the compounds would be beneficial.
- The conclusions provide a general pointer toward the possible application of labdanic diterpenoids as therapeutic agents for cancer. It would be helpful to provide specific suggestions for future study, e.g., structural modification aimed at enhancing the selectivity of the compounds towards cancer cells.
Author Response
- The authors describe the identification of the compounds by spectroscopic methods like NMR, IR, and MS. The spectra are not provided in the supplementary data, however, which would be helpful in verifying the results of identification. The inclusion of such information is necessary to ensure reproducibility and openness of results.
Response:
The supplementary data including NMR, IR and MS has been provided separately together with the manuscript.
- The figure 1 of the structures of the compounds isolated is of poor resolution (compound 1) and is hard to read.
Response:
The structures have been redrawn and inserted as a ChemDraw image for improved clarity and resolution as shown in Figure 2.
- Although the introduction is a good summary of why TNBC is so significant and why it's challenging to treat, the fact about the limited therapeutic options is restated. It would be better to have briefer presentation of this and clearer linkage to why this study is needed.
Response:
The discussion on limited therapeutic options has been condensed and the rationale for the study has been expanded.
- The cytotoxicity results are given in a table; however, preparatory analysis in the text lacks structure. Specifically, the relative assessment of the activity of the various compounds can be stated more clearly, highlighting the principal findings.
Response:
The relative activity of the various compounds has been clearly stated, with structured discussion highlighting the principal findings.
- Though the ADME properties of the compounds are analyzed by SwissADME, the discussion of the results remains relatively brief. A more in-depth exploration of how solubility and lipophilicity influence the potential pharmacological uses of the compounds would be beneficial.
Response:
A more in-depth exploration of how solubility and lipophilicity influence the potential pharmacological uses of the compounds has been included. References have also been provided for detailed information.
- The conclusions provide a general pointer toward the possible application of labdanic diterpenoids as therapeutic agents for cancer. It would be helpful to provide specific suggestions for future study, e.g., structural modification aimed at enhancing the selectivity of the compounds towards cancer cells.
Response:
Thank you for the comment. The conclusion now includes specific suggestions for future studies, such as structural modifications to enhance the selectivity of bis-spiro labdane toward cancer cells.
Round 2
Reviewer 1 Report
Comments and Suggestions for Authors
I recommend it to publication.
Reviewer 2 Report
Comments and Suggestions for Authors
Accept in present form
Reviewer 4 Report
Comments and Suggestions for Authors
The authors have revised the manuscript in accordance with the suggestions provided. I recommend the publication of the article.